# Well-Defined Thermo-Responsive Copolymers Based on Oligo(Ethylene Glycol) Methacrylate and Pentafluorostyrene for the Removal of Organic Dyes from Water

**DOI:** 10.3390/nano10091779

**Published:** 2020-09-08

**Authors:** Federica Zuppardi, Mario Malinconico, Franck D’Agosto, Giovanna Gomez D’Ayala, Pierfrancesco Cerruti

**Affiliations:** 1Institute for Polymers, Composites and Biomaterials (IPCB-CNR), 80078 Pozzuoli, Italy; fe.zuppardi@libero.it (F.Z.); mario.malinconico@ipcb.cnr.it (M.M.); cerruti@ipcb.cnr.it (P.C.); 2CPE Lyon, CNRS, UMR 5265, Chemistry, Catalysis, Polymers and Processes (C2P2), Université Claude Bernard Lyon 1, 69616 Villeurbanne, France; franck.dagosto@univ-lyon1.fr

**Keywords:** oligo(ethylene glycol) methacrylate, reversible addition-fragmentation chain transfer (RAFT), fluorinated thermo-responsive polymers, self-assembly, water remediation, organic dyes removal

## Abstract

Thermo-responsive copolymers based on oligo(ethylene glycol) methacrylate (OEGMA, *M_n_* = 300 g/mol) and pentafluorostyrene (PFS), coded PFG, were synthesized by RAFT polymerization, using a trithiocarbonate (CTTPC) as controlling agent. Different molar masses were targeted and dispersities lower than 1.51 were obtained. The thermally triggered self-assembly of the resulting PFG copolymers in water was investigated by dynamic light scattering (DLS). The lower critical solution temperature (*LCST*) slightly increased with the molecular weight in the 26–30 °C temperature range, whereas the sizes of the intermicellar aggregates formed upon self-assembly tended to decrease with increasing molecular weights (ranging from 1415 to 572 nm). The resulting thermally-induced polymer aggregates were then used to encapsulate and remove organic contaminants from water. Nile Red (NR) and Thiazole yellow G (TYG) were employed as hydrophobic and hydrophilic model contaminants, respectively. Experimental results evidenced that higher molecular weight copolymers removed up to 90% of NR from aqueous solution, corresponding to about 10 mg of dye per g of copolymer, regardless of NR concentration. The removal of TYG was lower with respect to NR, decreasing from about 40% to around 20% with TYG concentration. Finally, the copolymers were shown to be potentially recycled and reused in the treatment of contaminated water.

## 1. Introduction

Thermo-responsive polymers are stimuli-responsive macromolecules showing drastic and discontinuous changes of their physical properties with temperature [1,2,3,4]. In particular, they show solubility changes with temperature when dissolved in water. The consequent miscibility gaps in their temperature-composition diagrams result in the so-called lower critical solution temperature (*LCST*). Below this temperature, they are completely soluble in water, as the polymer chains exist in a random coil conformation thanks to the formation of hydrogen-bonding interactions between hydrophilic polymer moieties and surrounding water molecules. When the temperature exceeds the *LCST*, the hydrogen bonds are disrupted and hydrophobic interactions become dominant, leading to the collapse of polymer chains and the formation of aggregates. Macroscopically, a clouding of the solution occurs, and for this reason the *LCST* is also named clouding temperature (*T_c_*). The *LCST* value is determined by a balance between attractive polymer–water and polymer–polymer interactions. Therefore, the temperature is used as an external trigger to modulate and control the hydrophilicity, and therefore the shape of macromolecules. The capability to form nano- and micro-aggregates in diluted water solutions is extremely promising for many applications, including smart coatings, drug and gene delivery and water remediation [5,6,7,8,9]. As regards water purification, a big variety of contaminants can be found in industrial wastewater, and among them, synthetic dye molecules represent the major cause of pollution. In particular, azo dyes, mostly used in the textile industry, are of particular concern as they represent about 60% of all the reactive dyes and are potentially carcinogenic [10].

There are several methods, such as photoelectrocatalysis, electrocoagulation and biotransformation, which are effectively employed for removing organic pollutants from untreated wastewater. However, the use of thermo-responsive polymers is particularly attractive for water remediation applications. Indeed, the critical temperature at which the transition occurs is strictly dependent on the composition of copolymers and can be modulated depending on the desired application [11,12]. Increasing the temperature above *LCST* induces the formation of polymer aggregates, which are able to encapsulate guest molecules and can be easily separated from an aqueous solution by filtration or centrifugation. Several examples have been reported on the use of copolymers of N-isopropylacrylamide (NIPAAm) to remove pollutants from aqueous solutions, mostly dealing with the use of thermo-responsive hydrogels as adsorption assemblies [13]. In particular, the structures of these polymers have been designed according to affinities to specific dyes, as charged moieties are able to efficiently adsorb oppositely charged organic pollutants from water. Thus, several reports have investigated the absorption of anionic and cationic dyes by purpose-built charged hydrogels [10,14], while the use of nonionic hydrogels, in which hydrogen-bonding or hydrophobic interactions are responsible for dye removal, are considerably fewer [13]. Moreover, only a handful of papers has been exploiting the capability of aggregation of soluble thermo-responsive polymers to remove contaminants from water [15,16,17,18].

In the last few decades, amphiphilic copolymers based on oligo(ethylene glycol) methacrylate (OEGMA) have been largely investigated due their interesting thermo-responsive behavior [19,20,21,22,23,24,25,26,27,28,29]. POEGMA is composed of a methacrylate backbone and poly(ethylene glycol) branches. It shows a tunable *LCST* in the range of 30–90 °C, a reversible phase transition and bioinert properties. In OEGMA-based copolymers, the relative composition of hydrophilic and hydrophobic units strongly affects the *LCST*. In particular, an increase of the hydrophobic character leads to a decrease of *LCST*, while when the hydrophilicity increases, the phase transition occurs at higher temperatures. Therefore, the possibility of modulating the *LCST* by varying the copolymer composition makes OEGMA-based copolymers versatile and suitable for many applications. Fluorinated units are often incorporated in the copolymers to modulate the self-assembly process, as they have a strong tendency to segregate from water, due to their high hydrophobicity [30,31]. In a previous work, a series of thermo-responsive copolymers was prepared by free radical polymerization of OEGMA and pentafluorostyrene (PFS), and the effect of composition on self-assembly parameters was studied [12]. It was demonstrated that both *LCST* and aggregate size can be precisely tuned by modifying the copolymer composition. Moreover, several fluorinated OEGMA-based copolymers have been synthesized by controlled radical polymerization (CRP) methods [32,33,34,35,36,37,38]. CRPs usually allow one to prepare polymers with tunable molecular weights and chemical structures, and to prepare well-defined micelles via the design of new copolymers with controlled structures. Among CRPs, reversible addition-fragmentation chain transfer (RAFT) polymerization is promising due to its versatility for the synthesis of well-defined polymers with controlled molecular weight distributions and properties, and it is particularly compatible with different fluorinated monomers [12,34,38,39]. An accurate control of the copolymer structure, in terms of chemical composition and molar mass, would provide more control over the characteristics of self-assembled particles (*LCST* and hydrodynamic diameter) [34,39].

In this work, amphiphilic copolymers of OEGMA and PFS were synthesized with different molecular weights by RAFT, and their self-assembly behavior in water was studied. Furthermore, a preliminary evaluation of their potential as thermally triggered flocculants for water purification was carried out. In this respect, considering the amphiphilic nature of these copolymers, both hydrophobic and hydrophilic molecules were expected to potentially interact with the polymer backbone, through hydrophobic or polar interactions, respectively, Therefore, two azo dyes, Nile Red (NR) and Thiazole yellow G (TYG), were selected as model contaminants. NR, as an hydrophobic dye, was supposed to establish interactions with the apolar PFS moieties; Thiazole yellow G (TYG) is a hydrophilic dye potentially able to interact with the OEGMA poly(ethylene glycol) side chains through polar forces or hydrogen bonding.

## 2. Experimental

### 2.1. Chemicals

Pentafluorostyrene (PFS, 99%) and oligo(ethylene glycol) methyl ether methacrylate (OEGMA, *Mn* ~300 g/mol), provided by Sigma-Aldrich (St. Louis, MO, USA), were purified by passing over a neutral aluminium oxide column. Trioxane (99%), benzoyl peroxide (BPO, 98%), dimethylformamide (DMF, 99.8%), petroleum ether (ACS reagent) and tetrahydrofuran (THF, 99.9%) were from Sigma-Aldrich and used as received. 2-Cyano-2-propyl propyl trithiocabonate (CPPTC) was synthesized according to the procedure described by Chaduc et al. [40,41].

### 2.2. Synthesis of Copolymers of OEGMA and PFS (PFG) via RAFT

Copolymerization of OEGMA and PFS was performed in inert atmosphere, under RAFT conditions. In particular, 19.56 g (65.22 mmol) of OEGMA, 5.44 g (27.95 mmol) of PFS (corresponding to a 70/30 OEGMA/PFS molar ratio) and 0.28 g of trioxane were dissolved in 40 mL of dioxane. A solution of 0.11 g (0.5 mmol) of 2-cyano-2-propyl propyl trithiocabonate (CPPTC) used as a chain transfer agent (CTA) and 0.165 g (0.10 mmol) of azobisisobutyronitrile (AIBN) in 7 mL of dioxane was then added. The resulting solution was divided in several tubes, degassed with nitrogen for 15 min and placed in a preheated bath at 70 °C. During the polymerization, the tubes were taken at different reaction times and cooled in an ice bath, in order to recover polymers with different molecular weighst. The mixtures were dried at room temperature. Polymers were purified by dissolving them in THF and precipitating and washing twice in petroleum ether. Copolymers were dried under vacuum at room temperature until constant weight was reached.

The copolymers with different molecular weights obtained by stopping the reaction at predetermined times (3, 6, 8, 15 and 24 h) were coded as PFGX, where X stands for the reaction time in hours. The theoretical number-average molar mass of copolymers (*M_n theo_*) was calculated as follows:(1)$$M_{n\;theo}=\;\frac{\left({\left[OEGMA\right]}_{0}\times \;M_{OEGMA}+\;{\left[PFS\right]}_{0}\;\times \;M_{PFS}\right)}{{\left[CPPTC\right]}_{0}+2f{\left[I\right]}_{0\;}\left(1-e^{k_{d}t}\right)}\;\times \;\frac{\%wt_{conv}}{100}+\;M_{CPPTC}$$
where *M_OEGMA_*_,_
*M_PFS_* and *M_CPPTC_* are the molar masses of OEGMA, PFS and CPPTC; [OEGMA]_0_, [PFS]_0_, [CPPTC]_0_ and *[I]_0_* are the initial concentrations of the two monomers, CPPTC and AIBN respectively; and %*wt_conv_* is the weight conversion of monomers Equation (1). $2f{\left[I\right]}_{0\;}\left(1-e^{k_{d}t}\right)\;$ accounts for the contributions of initiator-derived chains, where *k_d_*_,_
*f* and *t* are the decomposition rate constant, the efficiency factor of the initiator and the polymerization time [42].

### 2.3. Polymer Characterization

^1^H-NMR spectroscopy analyses were performed to determine monomer conversion and the composition of the purified copolymers. ^1^H-NMR spectra were recorded in CDCl_3_ on a Bruker Avance III 400 MHz spectrometer (Milano, Italy). Spectra were acquired under the following measurements conditions: 90° pulse width 7.5 ms, relaxation delay 1 s, acquisition time 1.4 s and 16 scans.

Monomer conversion was calculated by relative integration of vinyl protons of OEGMA and PFS and protons of trioxane (used as an internal reference). The weighted average molar conversion was calculated as follows:(2)$$\%conv_{tot}=x_{OEGMA}\times \%conv_{OEGMA}+x_{PFS}\times \%conv_{PFS}$$
where *x_OEGMA_* and *x_PFS_* are the feeding molar fractions of OEGMA and PFS, respectively.

Size exclusion chromatography (SEC) was performed to determine molecular weight distributions, using a GPC Max Viscotek (Roma, Italy) system equipped with a TDA 305 triple detector, and three Phenomenex columns: precolumn, 106 and 103 g/mol. Samples were dissolved in THF (5 mg/mL) and the resulting solution was filtered through PTFE pore size 0.20 mm filters. THF was used as the eluent (0.8 mL/min) at 30 °C. All three detectors were calibrated with a PS standard having a narrow molecular weight distribution (*M_n_* = 115,000 g/mol, *Ð* = *M_w_*/*M_n_* = 1.02) provided by Viscotek.

### 2.4. Characterization of the Self-Assembly Properties

The self-assembly behavior of copolymers was investigated by dynamic light scattering (DLS). DLS measurements were performed with a Malvern Zetasizer Nano ZS instrument (Cambridge, UK) equipped with a 4 mV HeNe laser operating at *λ* = 633 nm, with a measurement angle of 173°. Mean size and polydispersity index were obtained by cumulants analysis of the correlation function.

For the determination of *LCST*s and the temperature–dependent sizes of polymer aggregates, copolymer solutions were analyzed at a concentration of 5 mg/mL, in a temperature range between 15 and 45 °C, with temperature increments of 5 °C. The experimental data were fitted with the Boltzmann equation (Equation (3)), to determine the values of characteristic parameters of the self-assembly process, and their dependence on the molecular weight:(3)Dh=Dh0−Dhf1+eT−LCSTdT+Dhf
where *D_h0_* and *D_hf_* are the equilibrium initial and final values of the hydrodynamic diameter, before and after the transition; *LCST* is the temperature at the curve inflection point, taken as the lower critical solution temperature; *dT* is related to the slope of the tangent line at *LCST*.

Dynamic light scattering (DLS) was utilized for the determination of *LCST* and the hydrodynamic diameter (*D_h_*) of the polymer aggregates in water as a function of temperature.

The morphologies of the micelles and their aggregates were evaluated by transmission electron microscopy (TEM). A TECNAI G12 Spirit-Twin (LaB6 source) microscope (Milano, Italy) equipped with a FEI Eagle 4k CCD camera (Milano, Italy) was used, operating with an acceleration voltage of 120 kV. Samples at *T* < *LCST* were prepared by casting a drop of copolymer aqueous solution (2.5 mg/mL) on a 300 mesh copper/carbon grid, which was allowed to dry at room temperature. For the imaging of samples above the *LCST*, the copolymer solutions were heated at *T* > *LCST* (60 °C). Concurrently, a TEM grid was equilibrated in an oven at the same temperature; then a drop of the heated solution was deposited on the grid and dried at 60 °C.

### 2.5. Dye Removal Analyses

The feasibility of using PFG copolymers for the treatment of water contaminated with organic pollutants was investigated. To that end, their ability to encapsulate organic dyes, used as model contaminants, was evaluated. The trials were carried out using two dyes: a hydrophobic one, Nile Red (NR), and a hydrophilic molecule, Thiazole Yellow G (TYG, also known as Titan yellow). In particular, the dye removal efficiency (*RE*) was evaluated by UV–Vis spectroscopy, as the percentage of dye removed above the *LCST*. Furthermore, removal capacity (*RC*) was determined as mg of dye removed per g of polymer. The effect of dye concentration on dye removal was investigated.

As concerns TYG, *RE* was determined on 5 mg/mL polymer solutions containing varying TYG amounts (0.01, 0.02 and 0.05 mg/mL). First, the absorbance value at 411 nm was determined at room temperature; then the solution was heated to 40 °C to trigger polymer self-assembly; and the turbid suspension was centrifuged at 40 °C using a HERMLE Centrifuge Z 32 HK (Wehingen, Germany). The residual TYG content in the supernatant was determined, and *RE* was calculated as *RE* = (*A_RT_* − *A_40_*)/*A_RT_* × 100, where *A_RT_* and *A_40_* were the absorbances of the solution at 411 nm, before and after centrifugation, respectively.

*RC* was determined as *RC* = 10 × *RE* × [TYG]/[PFGX], where [TYG] and [PFGX] were TYG and PFGX copolymer concentrations in mg/mL.

In the case of NR, both loading capacity (*LC*) and *RE* were evaluated. For *LC* determination, a 5 mg/mL PFG aqueous solution was prepared. NR was dissolved in ethyl acetate (10 mg/mL) and increasing amounts of the resulting solution were added to the polymer solution, until saturation. Upon each dye addition, the obtained mixtures were centrifuged at room temperature, and the content of NR in the supernatant was determined by UV–Vis spectroscopy at 552 nm, using a JASCO V-570 spectrophotometer (Easton, TX, USA). *LC* was determined as the NR concentration above which further dye additions did not cause any increase in the absorbance. *RE* and *RC* were determined following the procedure described above.

## 3. Results and Discussion

### 3.1. Syntheses and Chemical Characterizations of Copolymers of OEGMA and PFS

Thermo-responsive copolymers of OEGMA and PFS were synthesized in dioxane at 70 °C, in the presence of CPPTC used as the CTA and AIBN as the initiator (Scheme 1). The OEGMA/PFS molar ratio in the feed was fixed to 70/30, in order to obtain water-soluble copolymers, as it was noticed that PFS molar fractions exceeding 30% yielded water-insoluble copolymers [12]. The CPPTC to monomers (OEGMA + PFS) ratio was set to target a molecular weight (*M_n theo_*) of 50,000 g/mol. The kinetics of copolymerization were studied, and copolymers of different molecular weights were obtained after purification by stopping the reaction at different times. Each sample was analyzed by ^1^H-NMR and GPC, to evaluate monomer conversion and molecular weight characteristics, respectively. In particular, the monomer conversion was calculated by comparing the decreases of OEGMA (5.5 and 6.1 ppm) and PFS (5.7, 6.0 and 6.7 ppm) vinyl proton signals to the signal of the six protons of trioxane (5.1 ppm).

Figure 1a depicts individual and total molar conversion of monomers as a function of time. After the first 3 h of reaction, conversion values of PFS and OEGMA were 31% and 22%, respectively, pointing out that PFS reacted slightly faster than OEGMA. It was observed that PFS homopolymerization proceeds at a slow rate (around 15% of conversion after the first 3 h), as already reported for conventional free radical polymerization reaction [12]; this result suggests that PFS is preferably added to the growing chains, as PFS propagating radicals are probably more stable than those with OEGMA chain ends.

As the reaction further proceeded, conversion of both monomers steadily increased up to 15 h, with a slightly faster conversion rate ascribed to PFS. Subsequently, the conversion of PFS slowed down, achieving a value of around 60%, whereas OEGMA conversion steadily increased, up to an overall value of 56% after 24 h. The theoretical PFS to OEGMA molar ratio, calculated on the basis of monomer conversion (Table 1), was around 0.60 for all the copolymers, and it was significantly higher than that in the feed (0.43). A small decrease was noticed at 24 h, due to the slower PFS conversion rate.

Figure 1b shows the evolution of the SEC-THF chromatograms of the reaction mixtures quenched and analyzed at fixed times. The molecular weight distribution progressively shifted toward lower volumes with increasing conversion, characterizing the growth of the chains. For PFG15 and PFG24, a shoulder appeared at higher molecular weight, suggesting the presence of a bimodal distribution due to the occurrence of termination reactions.

The increase of the *M_n_* values plotted as a function of total monomer conversion (Figure 1c) confirms the control of the polymerization. Dispersity values stayed, rather, below 1.6 (Table 1). The theoretical PFS to OEGMA molar ratio in the chains, calculated on the basis of monomer conversion (Table 1), was around 0.60 for all the copolymers and it was significantly higher than that in the feed (0.43), showing a slight compositional drift taking place along the chains as a result of the control of the polymerization. Copolymers were purified and characterized through ^1^H-NMR spectroscopy in CDCl_3_. A representative spectrum of PFG24 showing the characteristic resonances of PFS and OEGMA protons is reported in Figure 1d. In particular, the signals of pendant PEG moieties at 3.4 ppm, 3.6 ppm and 4.1 ppm relative to protons of methoxy (–OCH_3_), main chain methylenes (–O–CH_2_–CH_2_–) and methylene next to ester groups (–CH_2_–O–C=O–), respectively, can be clearly seen. Furthermore, peaks at 1.4–2 ppm and 0.7–1.2 ppm were due to methacrylic backbone –CH_2_– and –CH_3_ protons of OEGMA domains, respectively. Finally, signals in the region 2.0–3.1 ppm were attributed to backbone protons of the PFS units.

The molar composition of copolymers was calculated from the ratio between PFS (2.0–2.8 ppm) and OEGMA (3.4 ppm) methyl resonance integrations (Table 1). PFS to OEGMA molar ratios of purified copolymers were roughly constant at different molecular weights. Moreover, they were in agreement with the values calculated on the basis of monomer conversion (PFS/OEGMA_theo_). Slight differences are evidenced only for low and high molecular weight copolymers (PFG3 and PFG24, respectively). In particular, at low molecular weight, the ratio of PFS/OEGMA was lower than theoretical one. These results suggest that short chains, richer in PFS, are eliminated during the purification process. This outcome is in agreement with the observed higher conversion rate of PFS at early stages of polymerization.

On the other hand, at high molecular weight, PFS/OEGMA was higher than the theoretical ratio, as in the late stages of polymerization side reactions involved essentially OEGMA. The number-average molecular weights of the synthesized copolymers were in good agreement with the theoretical values up to 8 h of reaction (Table 1). This demonstrates that within this time interval the number of chains is governed by the RAFT agent concentration, suggesting a high apparent chain transfer constant of the RAFT agent in the polymerization system [43]. Subsequently, an upward deviation from the theoretical value was noticed, showing that at a later stage reaction control is partly lost, as also confirmed by the polydispersity (***Ð***) value increase.

To summarize, the obtained results hint towards a loss of control in the reacting system. Indeed, in the initial stage of polymerization polymer chains grow in a controlled way. On the other hand, as the reaction proceeds at 70 °C, the number of AIBN-initiated chains increases due to the larger number of AIBN radicals. In this way, the termination reactions become more significant, leading to Mn values higher than expected [44].

### 3.2. Self-Assembly of PFG Copolymers in Aqueous Solution

Copolymers of OEGMA and PFS are soluble in water at low temperature, thanks to the formation of hydrogen bonds between hydrophilic groups and surrounding water molecules. However, due to the copolymer amphiphilic nature, these bonds are counterbalanced by unfavorable interactions between apolar regions and water, leading to the formation of micelles [12]. A micellar structure, with a hydrophobic core surrounded by an external hydrophilic shell, was confirmed by the comparison of NMR spectra of PFG recorded in CDCl_3_ and D_2_O (Figure 2a). In chloroform, the signals relative to all OEGMA and PFS protons appeared sharp and intense, indicating that copolymer chains were completely dissolved as unimers. A significant change was noticed when the spectrum was recorded in deuterium oxide. Indeed, signals relative to PFS protons (2.0 to 2.8 ppm) completely disappeared, while peaks corresponding to OEGMA protons decreased in intensity and broadened, due to the reduced mobility of OEGMA backbone [38]. Furthermore, the signals corresponding to OEGMA side chains, in particular, relative to the methoxy protons, were relatively sharp and shifted to lower chemical shifts, suggesting that these groups were still in contact with the surrounding water molecules and maintained significant mobility [45,46]. It is worth underlining that the above-mentioned compositional drift may contribute to the micelle formation, via the gradient-like structure of the chains, richer in OEGMA upon their growth during polymerization.

Figure 2b displays the scattering intensity distribution as a function of hydrodynamic diameter for a 5 mg/mL PFG24 aqueous solution, upon heating from 15 to 45 °C. A monomodal distribution with a roughly constant polydispersity (from 0.14 to 0.18) was observed at each temperature. Up to 30 °C, the polymer solution appeared transparent and 10–15 nm sized micelles were detected, as shown also by TEM observation (Figure 2c). Above this temperature, the particles grew in size until 600 nm, due to micelle aggregation, conferring turbidity to the solution. TEM observation confirmed the presence of roughly round-shaped particles having a quite large size distribution. Figure 2d summarizes the temperature-dependent change in *D_h_*. For all copolymers, at temperatures well below the *LCST*, the size of the polymer supramolecular structures was about 10 nm, irrespective of copolymer molecular weight. On heating, all samples exhibited a sudden increase of *D_h_* due to polymer micelle aggregation. To precisely evaluate *LCST*, DLS measurements were performed at a heating rate of 1 °C/min and the experimental data were fitted to a Boltzmann equation [12]. The resulting *LCST* values are listed in Table 1. It is noted that clouding occurs in the 26–30 °C temperature range, and the *LCST* slightly increased with the molecular weight. Further, as a general trend, the size of the intermicellar aggregates tended to decrease with increasing molecular weights (Table 1). As an example, at 40 °C *D_h_* dropped down from 1415 nm of PFG6 to 572 nm of PFG24. An exception to this trend is PFG3, which forms smaller aggregates. Likely, the low molecular weight and the low ratio PFS to OEGMA enhance polymer solubility in water at high temperature, promoting the formation of smaller particles. In addition, the size of polymer aggregates showed slight fluctuations on heating above the *LCST*, indicating that the aggregating particles partially dissociated and restructured in response to small changes in the balance between polymer–polymer and polymer–water interactions [12]. This behavior was mainly observed for the lower molecular weight polymers. On the other hand, PFG24 showed a more regular trend of aggregate size with the temperature, as already reported for higher molecular weight OEGMA/PFS copolymers synthesized by conventional radical polymerization (PFGFRP, Mn between 56 and 89 kDa) [12]. That experimental evidence suggests that, for the copolymers with lower molecular weight, the hydrophilic side chains and the RAFT agent bound to chain ends affect more significantly aggregate size and stability with respect to PFG24, in which several chains are not even ended with the RAFT agent, because of the occurrence of termination reactions.

A similar behavior has been described for PNIPAAm, for which high molecular weight chains give sharp cloud point transitions, whereas those having lower polymerization degree show broader transitions [15,47]. It is worth highlighting that upon self-assembly in water, the copolymers prepared by RAFT polymerization (PFG_RAFT_) form aggregates larger than those observed for PFG_FRP_. As an example, the corresponding PFG copolymer synthesized by free radical copolymerization, PFG_FRP_, showed a D_h_ of about 80 nm [12]. This evidence is probably due to the higher hydrophobic character of the PFG_RAFT_ with respect to the PFG_FRP_ copolymers (PFS/OEGMA in PFG_RAFT_ = 0.50–0.60; PFS/OEGMA in PFG_FRP_ = 0.42). The higher content of PFS moieties is probably responsible for a stronger tendency to self-assembly, which leads to formation of larger aggregates. A comparison of the NMR spectra recorded at room temperature in water (Figure 3) evidences that the OEGMA side chains signals of PFG24 are broader and slightly downfield shifted compared to PFG_FRP_.

These results suggest that OEGMA side chains of PFG24 are less solvated in water, and likely more involved in intermolecular interactions within the polymer aggregates, eventually yielding larger aggregates above the *LCST*. However, a more accurate comparison between the self-assembly properties of PFG_FRP_ and PFG_RAFT_ would require the synthesis of copolymers with similar molecular weights and PFS/OEGMA ratios, which is behind the scope of the present paper and will be the object of further studies.

The micrometric size of PFG copolymer aggregates makes them suitable for applications in which thermally triggered self-assembly is required, such as flocculation for pollutant removal, as described in the next section.

### 3.3. Evaluation of PFG Copolymers for Water Purification

As seen in previous sections, PFG copolymers have the ability to form self-assembled aggregates by increasing the solution temperature. The self-aggregation and the ability to encapsulate different molecules can be exploited to efficiently remove pollutants from contaminated water. In this section, the study of the copolymers’ removal efficiencies of dyes in aqueous solutions are reported.

A large variety of organic and inorganic contaminants is present in industrial wastewater. Among them, both poorly and highly soluble organic dyes have been proven to have major impacts on water pollution [16]. Therefore, the ability of PFG copolymers to remove organic dyes from water was evaluated. To that end, preliminary tests were carried out using two dyes as model compounds: a hydrophobic dye, Nile Red (NR), and a hydrophilic molecule, Thiazole Yellow G (TYG) (Scheme 2). Three polymers were selected for this analysis, namely, PFG8, PFG15 and PFG24, since they showed higher *LCST*s and molecular weights (Table 1).

When NR was used as a model compound, the study of the PFG pollutant removal ability included two steps, namely, the evaluation of both *LC* and *RE*. *LC* represents the capability of amphiphilic PFG copolymers to solubilize a poorly soluble dye at room temperature. Indeed, in water NR tends to aggregate, showing a maximum solubility of about 0.003 mg/mL [48]. To determine *LC*, increasing amounts of a NR solution were added to an aqueous polymer solution, and the resulting mixture was centrifuged at room temperature to remove remaining non-soluble NR. The polymer-encapsulated NR in the supernatant was then quantified by UV spectroscopy [16]. As an example, UV–Vis spectra of PFG8 aqueous solutions (5 mg/mL) containing varying concentrations of NR are shown in Figure 4a.

The absorbance values gradually increased with the dye concentration, until a maximum value was reached, corresponding to the *LC* (0.05 mg/mL). Subsequent addition of NR resulting in no further increase in absorbance showed that the additional dye only precipitates. This result entails that PFG copolymers (5 mg/mL) are able to increase NR solubility by more than ten times.

Figure 4b shows the absorbance value at 552 nm as functions of NR concentrations for the three PFG copolymers. The increasing trend of absorbance was similar, irrespective of the copolymer molecular weights, and *LC* turned to be around 0.05 mg/mL for all the tested copolymers. Once the *LC* was evaluated, the solutions were centrifuged at a temperature above *LCST* (40 °C) to induce polymer self-assembly and allow separation of the NR-loaded polymer aggregates from the solution. *RE* was then determined by UV–Vis spectroscopy, as the ratio of absorbance values after and before centrifugation. The obtained results are reported in Figure 5a, which summarizes the RE values as a function of NR concentration for the PFG thermo-responsive copolymers.

It was noticed that higher molecular weight copolymers (PFG15 and PFG24) exhibited RE values above 90%, regardless of NR concentration, whereas *RE* significantly depended on NR concentration in the case of PFG8. In this case, indeed, *RE* increased with NR concentration from 58% (0.01 mg/mL) to 89% (0.05 mg/mL).

The mechanism of NR removal mainly depends on the hydrophobic interactions between the alkyl and aromatic moieties of NR and hydrocarbon parts of the PFG copolymers [15]. Additionally, successful removal also requires that polymer self-aggregation results in the formation of particles sufficiently large to be recovered by centrifugation. It is hypothesized that for the PFG copolymer a critical concentration exists below which the presence of NR interferes with the formation of stable intermolecular interactions mediated by the polymer hydrophobic moieties. In this way, chain collapse and formation of aggregates could be hindered, thereby reducing the contaminant removal capability. In general, the *RE* values obtained herein are generally higher than those reported for thermo-responsive macro- and micro-hydrogels based on PNIPAAm, for which *RE* values ranging from 20% to about 90% have been reported [10,13,14]. However, a more comprehensive study of the concentration-dependent self-assembly process in presence of different dyes will be tackled in a forthcoming work.

Figure 5b shows *RC*, expressed as mg of dye removed per g of polymer, as a function of NR concentration. For all copolymers, a linear increase of the amount of dye removed was observed, with PFG8 exhibiting lower *RC* values, particularly at low NR concentrations. As a matter of fact, removal capacity is expected to increase with an increase of the concentration of dyes, and to reach an equilibrium plateau value depending on the fact that the number of adsorption sites for the dye on polymer chains are finite, so that no more dye molecules can be further removed [49]. The linear trend observed for PFG copolymers suggests that in the explored range of NR concentration (dictated by the poor water solubility of NR) the theoretical maximum *RC* was not achieved yet, since further NR addition would result in dye precipitation. In these conditions, the experimental removal capacity showed a maximum value of about 10 mg/g at 0.05 mg/mL NR concentration. A comparison with the very few reported literature instances of NR removal with thermo-responsive polymers revealed that *RC* determined with PFG copolymers was significantly higher than that obtained using PNIPAAm [13], but lower than that reported using 2-hydroxy-3-butoxypropyl hydroxyethyl cellulose (HBPEC) [16]. In the latter case, the inferior *RE* was essentially due to the lower NR loading capacity.

As concerning water decontamination applications, thermo-responsive polymers, and specifically those based on PNIPAAm, have been employed essentially to remove ionic organic dyes [10,13,14,50]. Therefore, taking into account the amphiphilic character of PFG copolymers, a preliminary investigation on the ability of the latter to remove hydrophilic molecules was carried out. Thiazole Yellow G (TYG) was selected, which is an aromatic anionic dye, bearing also a secondary amino group, able to interact with hydrophobic moieties and to form ion association or hydrogen bonds with charged molecules or surfactants, depending on the solution pH [51,52].

Therefore, the mechanism of TYG removal depends on the formation of hydrophobic and electrostatic interactions between the dye molecules and the polymer chains. The evaluation of the TYG removal was performed in the same conditions used for NR, without adjusting the pH of the solution, in order to compare the removal capability of TYG with that observed in the presence of NR.

Figure 6a shows the RE values for TYG as a function of dye concentration. For all copolymers, RE decreased with TYG concentration from about 40% (0.01 mg/mL) to around 20% (0.05 mg/mL). At the intermediate concentration, PFG8 exhibited a higher *RE* (37%) compared to PFG15 and PFG24 (about 25%). Overall, TYG *RE* values were significantly lower than those determined for NR, especially at higher dye concentrations. Such a behavior demonstrates that, in order to completely remove TYG, both hydrophobic and electrostatic interactions are needed [53,54], the latter being not possible with non-charged PFG copolymers. Figure 6b displays the RC changes over the explored TYG concentration range. The removal capacities of PFG copolymers increased with increases in the initial dye concentrations, reaching values ranging from 2.0 to 2.5 mg/g, although no significant differences were observed for the different copolymers. This evidence highlights that increasing concentrations of TYG favor the self-assembly process, resulting in a higher *RC*. Indeed, it is well known that salts or buffering agents in solution tend to favor the salting out of OEGMA-based polymers [38,55], and to promote aggregation of the precipitated polymer assemblies [15].

In a final experiment, the potential recyclability of the centrifuged polymer, and its reusability for treating the same contaminated water sample, in order to improve the overall removal efficiency, were probed. In particular, after the first removal process, the TYG-loaded copolymer was separated from the solution containing the non-removed dye. The recovered polymer was purified from the loaded dye and then reused to remove further TYG from the dye containing solution. More specifically, after centrifugation at 40 °C of 4 mL of a PFG8 solution containing 0.01 mg/mL of TYG (first removal cycle), the precipitate made of PFG8 copolymer and encapsulated TYG (about 20 mg) was recovered and re-suspended with 3 mL of diethyl ether. The copolymer was soluble in this solvent, whereas TYG was insoluble, and therefore it was separated by centrifugation (10,000 rpm). The polymer-containing supernatant was then isolated and evaporated to dryness, and the recovered polymer was further used to treat the solution containing the TYG not removed after the first removal treatment. As can be seen in Figure 6c, while the first treatment resulted in a *RE* of 36%, the second treatment resulted in a *RE* of 25%. The main reasons for the RE decrease of the recycled copolymer were the incomplete recovery of copolymer after centrifugation and the presence of small amounts of dye in the recovered copolymer. Indeed, gravimetric measurements demonstrated that about 95% of PFG8 was recovered after centrifugation, while UV–Vis spectroscopy showed that about 5% of TYG was still present in the recovered copolymer. Nonetheless, an overall 52% TYG removal was obtained after two treatment cycles. This result suggested that PFG copolymers can be potentially recycled and reused in order to increase the purity extent of a contaminated water solution.

## 4. Conclusions

Amphiphilic thermo-responsive copolymers of OEGMA and PFS were synthesized under RAFT conditions, and their self-assembly behavior in aqueous solutions was studied. Moreover, preliminary experiments were carried out to test their potential use for water remediation applications.

RAFT polymerization was carried out in dioxane at 70 °C in the presence of CPPTC as a RAFT agent and AIBN as an initiator. Copolymers at different molecular weights were obtained by stopping the reaction at fixed times and purifying the obtained products. For each synthesis, monomer conversion and copolymer composition were evaluated by NMR spectroscopy. Evaluation of monomer conversion revealed that at beginning of the reaction PFS reacted faster than OEGMA, while as the polymerization proceeded, PFS conversion slowed down and OEGMA conversion steadily increased. 

PFS to OEGMA molar ratios of purified copolymers were roughly constant at different molecular weights, but Mn linearly increased with conversion, suggesting that copolymerization was well controlled, according to the shift of molecular weights and the low dispersity values. The self-assembly behavior in water was investigated by DLS. At room temperature, all copolymers formed micelles of about 10 nm, irrespective of copolymer molecular weight. On heating, polymer micelle aggregated above the *LCST*, yielding micron-sized particles. The *LCST* slightly increased with the molecular weight, ranging from 26 to 30 °C, whereas the size of the inter-micellar aggregates tended to decrease with increasing molecular weights from 1415 to 572 nm.

A preliminary characterization of the applicability of these copolymers to remove organic dyes from water was carried out. Nile Red (NR) and Thiazole yellow G (TYG) were employed as hydrophobic and hydrophilic model molecules respectively. A loading capacity of 0.05 mg/mL was found for NR, with removal efficiencies above 90%, corresponding to 10 mg of dye per gram of polymer. As regards TYG, removal efficiency decreased with TYG concentration from about 40% (0.01 mg/mL) to around 20% (0.05 mg/mL). Finally, recyclability of copolymer was evaluated, demonstrating that PFG copolymers can be potentially recycled and reused in the treatment of contaminated water.

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
