# Peer review of "Well-Defined Thermo-Responsive Copolymers Based on Oligo(Ethylene Glycol) Methacrylate and Pentafluorostyrene for the Removal of Organic Dyes from Water"

_nanomaterials, 2020, doi:10.3390/nano10091779_

Round 1
Reviewer 1 Report
The manuscript from Gomez d'Ayala and coworkers describes the synthesis of amphiphilic and thermo-responsive copolymers. Upon triggering, the copolymers can trap model water polluants with relatively good efficiency. The manuscript is well written and easy to follow, whilst the discussion is sound. One can only have some comments and suggestions.
P3-L106 Please change "cahin" to chain. Could the authors provide a reaction scheme (even one relatively similar to the one in ref 11) ?
P3-L116 and P6. The choice of temperature for the current RAFT polymerization initiated with AIBN (10h half life @ 65 C) can result in higher amounts of initiator derived chains than what the authors consider herein. Although this proportion would be low, it is surprising that the authors do not consider this in their calculation of theoretical molar masses. Besides, the contribution of those chains may help the authors in their discussions in pages 7 on the amount of terminated chains and the discussion in page 9 (L311-313). In addition to the calculation, it is surprising that the authors do not display the theoretical evolution of molar masses with increasing conversion in Figure 1C. Have the authors tried to measure molar masses by NMR ?
P3 On the discussion about radical-radical termination, could the authors show the full NMR spectrum in Figure 1D ?
P9-L330 Perhaps this experiment is necessary here. In the current state, one cannot clearly understand if the present copolymer design is beneficial due to either the presence of the RAFT agent, or the low molar masses or the lower dispersity values.
P12-L439 One fails to understand the "recycling" experiment. Have the authors also checked the presence of the dye upon drying ? Could the authors compare the efficiency of a "recycled" copolymer and a "new" copolymer?
Reviewer 2 Report
In the manuscript entitled “Well-Defined Thermo-Responsive Copolymers Based on Oligo(Ethylene Glycol) Methacrylate and Pentafluorostyrene for the Removal of Organic Dyes from Water”, Gomez d’Ayala and co-workers reported the preparation of a series of copolymers PFG for the removal dyes from aqueous environment. The preparation and characterization of the thermo-responsive copolymers were well presented and discussed. However, the design of dye removal experiment is confusing. Therefore, I suggest its major revision before further consideration in Nanomaterials.
Specifically,
- The selection of copolymers and dyes is confusing. To achieve optimal dye removal, the structure of copolymers should be designed according to the affinity with specific dyes. Therefore, it is difficult to expect good dye removal efficiencies multiple dyes from one single copolymer. Does the use of PFS show any advantage of than other hydrophobic units, like polystyrene? For the removal of TYG, will the authors observe higher efficiency if they incorporate some cationic units onto their copolymers?
- Did the copolymer facilitate the dissolving of dyes? What is the fluorescent intensity of NR in water without any polymers? If there is no significant differences between the solubility of NR in the presence and absence of PFG, the term “loading capacity” is very confusing as the dye is not “loaded” on the polymers.
- The aqueous pH is an important parameter to change the solubility and fluorescence of dyes. It is not controlled, or at least not listed or discussed in the manuscript.
- The successful re-use of PFG to remove TYG is somehow expected due to the low affinity between the two molecules. Is the polymer still re-usable after removal of NR? If there is no simple methods to isolate PFG from high-affinity dyes, I doubt the efficiency and cost to re-use the polymers.
Round 2
Reviewer 1 Report
The authors are thanked for their efforts in revising this manuscript. It is suitable for publication.
Reviewer 2 Report
The authors have made significant improvements of the manuscript. Congratulations!
I look forward to seeing the new results with the new results of the terpolymer containing DMAEA.